# Age and the association between apolipoprotein E genotype and Alzheimer disease: A cerebrospinal fluid biomarker–based case–control study

Hana Saddiki[1], Aurore Fayosse[1], Emmanuel Cognat[2], Séverine Sabia[1], Sebastiaan Engelborghs[3,4], David Wallon[5], Panagiotis Alexopoulos[6], Kaj Blennow[7,8], Henrik Zetterberg[7,8,9], Lucilla Parnetti[10], Inga Zerr[11], Peter Hermann[11], Audrey Gabelle[12], Mercè Boada[13], Adelina Orellana[13], Itziar de Rojas[13], Matthieu Lilamand[2], Maria Bjerke[14], Christine Van Broeckhoven[14], Lucia Farotti[10], Nicola Salvadori[10], Janine Diehl-Schmid[6], Timo Grimmer[6], Claire Hourregue[2], Aline Dugravot[1], Gaël Nicolas[5], Jean-Louis Laplanche[15], Sylvain Lehmann[16], Elodie Bouaziz-Amar[15], the Alzheimer's Disease Neuroimaging Initiative[1], Jacques Hugon[2], Christophe Tzourio[17], Archana Singh-Manoux[1,18], Claire Paquet[2], Julien Dumurgier[1,2]*

1 Université de Paris, Inserm U1153, Epidemiology of Ageing and Neurodegenerative diseases, Paris, France, 2 Cognitive Neurology Center, Lariboisiere—Fernand Widal Hospital, AP-HP, Université de Paris, Paris, France, 3 Department of Neurology, Universitair Ziekenhuis Brussel, Center for Neurosciences, Vrije Universiteit Brussel, Brussels, Belgium, 4 Department of Biomedical Sciences, Institute Born-Bunge, University of Antwerp, Antwerp, Belgium, 5 Inserm U1245, Rouen University Hospital, Department of Neurology and CNR-MAJ, Normandy Center for Genomic and Personalized Medicine, Rouen, France, 6 Department of Psychiatry and Psychotherapy, Klinikum rechts der Isar, Faculty of Medicine, Technical University of Munich, Munich, Germany, 7 Department of Psychiatry and Neurochemistry, University of Gothenburg, Mölndal, Sweden, 8 Clinical Neurochemistry Laboratory, Sahlgrenska University Hospital, Mölndal, Sweden, 9 Department of Neurodegenerative Disease, Institute of Neurology, University College London, UK Dementia Research Institute, London, United Kingdom, 10 Center for Memory Disturbances-Lab of Clinical Neurochemistry, Section of Neurology, University of Perugia, Italy, 11 Department of Neurology, Clinical Dementia Center, University Medical Center Göttingen and German Center for Neurodegenerative Diseases, Göttingen, Germany, 12 Department of Neurology, Memory Research and Resources Centre, University of Montpellier, Montpellier, France, 13 Research Center and Memory Clinic, Fundació ACE, Institut Català de Neurciències Aplicades, Universitat International de Catalunya, Barcelona, Spain, 14 VIB Center for Molecular Neurology, Institute Born-Bunge and Department of Biomedical Sciences, University of Antwerp, Antwerp, Belgium, 15 Department of Biochemistry and Molecular Biology, Lariboisière Hospital, APHP, Paris, France, 16 Department of Biochemistry, University of Montpellier, Montpellier, France, 17 Bordeaux Population Health Research Center, Team HEALTHY, UMR1219, University of Bordeaux, Inserm, Bordeaux, France, 18 Department of Epidemiology and Public Health, University College London, London, United Kingdom

* julien.dumurgier@inserm.fr

**Data Availability Statement:** Data coming from the European memory clinics used in this study are freely available at: https://figshare.com/articles/

## Abstract

### Background

The ε4 allele of apolipoprotein E (*APOE*) gene and increasing age are two of the most important known risk factors for developing Alzheimer disease (AD). The diagnosis of AD based on clinical symptoms alone is known to have poor specificity; recently developed diagnostic criteria based on biomarkers that reflect underlying AD neuropathology allow better

CSF_biomarkers_data_pdf/12030084. The Whitehall-II Study is managed by the Department of Epidemiology and Public Health of the University College of London (UCL). The list of the principal investigators of the study can be shown here: https://www.ucl.ac.uk/epidemiology-health-care/research/epidemiology-and-public-health/research/whitehall-ii/people/principal-investigators. Data coming from the Whitehall-II Study can be made freely available to interested researchers upon request: https://www.ucl.ac.uk/epidemiology-health-care/research/epidemiology-and-public-health/research/whitehall-ii/data-sharing. Contact information: whitehall2@ucl.ac.uk. The Alzheimer's Disease Neuroimaging Initiative (ADNI) is a longitudinal multicenter study designed to develop clinical, imaging, genetic, and biochemical biomarkers for the early detection and tracking of Alzheimers disease. The principal investigator is Pr Michael W. Weiner, Professor of Radiology, Medicine, Psychiatry, and Neurology at the University of California San Francisco. Data coming from the ADNI study can be made freely available to interested researchers upon request: http://adni.loni.usc.edu/data-samples/access-data/. Contact information: adni-study@usc.edu. The 3-City Study is managed by the INSERM Unit U708, Université Victor Segalen, Bordeaux, France. The principal investigators are Pr Jean-François Dartigues and Pr Christophe Tzourio. Data coming from the 3-City Study can be made freely available to interested researchers upon request: http://www.three-city-study.com/the-three-city-study.php. Contact information: E3C.U708@inserm.fr.

**Funding:** The authors received no specific funding for this work.

**Competing interests:** No conflicts of interest with regards to this paper. Outside the submitted work, TG reported having received consulting fees from Actelion, Biogen, Eli Lilly, Iqvia/ Quintiles; MSD; Novartis, Quintiles, Roche Pharma, lecture fees from Biogen, Lilly, Parexel, Roche Pharma, and grants to his institution from Actelion and PreDemTech.

**Abbreviations:** Aβ, β-amyloid; *ABCA7*, ATP-binding cassette, sub-family A, member 7; AD, Alzheimer disease; ADNI, American Alzheimer's Disease Neuroimaging Initiative; *APOE*, apolipoprotein E; *APP*, amyloid precursor protein gene; A/T/N, β-amyloid deposition, pathologic tau, neurodegeneration; AUC, area under ROC curve; CI, confidence interval; CSF, cerebrospinal fluid; MMSE, Mini-Mental Status Examination; OR, odds ratio; PAF, population attributable fraction; *PSEN1*, presenilin-1 gene; *PSEN2*, presenilin-2 gene; p-Tau 181, tau phosphorylated at threonine 181; ROC,

assessment of the strength of the associations of risk factors with AD. Accordingly, we examined the global and age-specific association between *APOE* genotype and AD by using the A/T/N classification, relying on the cerebrospinal fluid (CSF) levels of β-amyloid peptide (A, β-amyloid deposition), phosphorylated tau (T, pathologic tau), and total tau (N, neurodegeneration) to identify patients with AD.

## Methods and findings

This case–control study included 1,593 white AD cases (55.4% women; mean age 72.8 [range = 44–96] years) with abnormal values of CSF biomarkers from nine European memory clinics and the American Alzheimer's Disease Neuroimaging Initiative (ADNI) study. A total of 11,723 dementia-free controls (47.1% women; mean age 65.6 [range = 44–94] years) were drawn from two longitudinal cohort studies (Whitehall II and Three-City), in which incident cases of dementia over the follow-up were excluded from the control population. Odds ratio (OR) and population attributable fraction (PAF) for AD associated with *APOE* genotypes were determined, overall and by 5-year age categories. In total, 63.4% of patients with AD and 22.6% of population controls carried at least one *APOE* ε4 allele. Compared with non-ε4 carriers, heterozygous ε4 carriers had a 4.6 (95% confidence interval 4.1–5.2; $p < 0.001$) and ε4/ε4 homozygotes a 25.4 (20.4–31.2; $p < 0.001$) higher OR of AD in unadjusted analysis. This association was modified by age ($p$ for interaction < 0.001). The PAF associated with carrying at least one ε4 allele was greatest in the 65–70 age group (69.7%) and weaker before 55 years (14.2%) and after 85 years (22.6%). The protective effect of *APOE* ε2 allele for AD was unaffected by age. Main study limitations are that analyses were based on white individuals and AD cases were drawn from memory centers, which may not be representative of the general population of patients with AD.

## Conclusions

In this study, we found that AD diagnosis based on biomarkers was associated with APOE ε4 carrier status, with a higher OR than previously reported from studies based on only clinical AD criteria. This association differs according to age, with the strongest effect at 65–70 years. These findings highlight the need for early interventions for dementia prevention to mitigate the effect of *APOE* ε4 at the population level.

### Author summary

#### Why was this study done?

- The ε4 allele of apolipoprotein E (*APOE*) gene (*APOE4*) and increasing age are two of the most important known risk factors for developing Alzheimer disease (AD).

- The recent development of diagnostic criteria based on biomarkers that reflect brain β-amyloid and tau lesions (β-amyloid deposition, pathologic tau, neurodegeneration [A/T/N] classification]) increases homogeneity in diagnosed cases.

- The strength of association of AD with risk factors can be better determined using biomarker-based AD compared with AD diagnosis based only on clinical criteria because

receiver operating characteristic; *SORL1*, sortilin-
related receptor; STROBE, Strengthening the
Reporting of Observational Studies in
Epidemiology; *TREM2*, triggering receptor
expressed on myeloid cells-2.

the latter are known to lack specificity as a result of difficulties in ruling out other causes
of dementia.

## What did the researchers do and find?

- We compared the overall and age-specific association between *APOE4* and AD using a
  case–control study that included 1,593 AD cases from memory clinics with positive
  cerebrospinal fluid biomarkers and 11,723 dementia-free controls drawn from two lon-
  gitudinal cohort studies.

- The use of a large number of cases and controls allows assessment of whether the associ-
  ation between *APOE4* and AD is dependent on age.

- Compared with controls, patients with AD were more likely to carry one *APOE4* (odds
  ratio [OR] = 4.6) or two *APOE4* (OR = 25.3). This association was significantly modified
  by age, with the strongest association seen between 65 and 70 years of age and weaker
  associations at the two tails of the age distribution.

## What do these findings mean?

- Incorporating biomarkers for diagnosis of AD identified an association with *APOE4*
  that is apparently greater than has been previously reported using clinical diagnosis of
  the disease.

- The impact of *APOE4* on the risk of AD was strongest between the 65 and 70 years of
  age, earlier than the mean age at diagnosis in this study, which was 72.8 years.

## Introduction

Apolipoprotein E (*APOE*) ε4 allele is the strongest known genetic risk factor for Alzheimer
disease (AD) in the general population [1]. The three most common alleles of the *APOE* gene
are ε2, ε3, and ε4; they encode for three isoform proteins differing in two single cysteine-to-
arginine amino acid substitution at positions 112 and 158 [2]. The ε3 allele is the most com-
mon allele in the population and is used as the reference to estimate risk of AD. Current evi-
dence suggests that ε4 heterozygote carriers have an overall 3-fold increased risk of AD,
whereas ε4 homozygote carriers have up to a 15-fold increased risk [3]. Conversely, the ε2
allele is associated with nearly 50% lower risk of AD [4]. The mechanisms underlying the rela-
tionship between *APOE* ε4 and AD are thought to be complex [5], involving β-amyloid (Aβ)
peptide clearance [6] as well as a direct role on neuronal death [7,8] and on phosphorylation of
tau [9].

Beyond individual genetic susceptibilities, increasing age is the main risk factor of AD [10].
Rare before the age of 60, AD affects up to 20% of the population after 80 years [11]. Accord-
ingly, the objective of the present study was to examine whether age modifies the association
between *APOE* genotype and risk of AD. Indeed, much of the evidence in this domain is from
studies undertaken before the 2000s, when the diagnosis of AD was based solely on clinical

criteria. Diagnosis of AD based on clinical criteria is marked by poor overall specificity, ranging from 40% to 70% compared with neuropathologic examination [12], which can lead to misclassification bias in studies [13]. Furthermore, the rate of false-positive diagnosis of AD may be influenced by the *APOE* status and age of patients, as the proportion of some differential diagnoses, like frontotemporal dementia, is higher in younger patients, leading to differential misclassification bias and unpredictable effect when estimating the strength of associations in case–controls studies [14].

Tau and Aβ peptide biomarkers are now incorporated in the new research diagnostic criteria in order to increase the biological homogeneity in diagnosed AD cases [15,16], and the β-amyloid deposition, pathologic tau, neurodegeneration (A/T/N) classification has recently been proposed as an unbiased biological definition of the disease [17]. Therefore, the present study aims to examine the global and age-specific association of APOE status with AD risk using a case–control design based on data on AD cases from nine European memory centers and the American Alzheimer's Disease Neuroimaging Initiative (ADNI) study [18], in which diagnosis were based both on clinical criteria [16] and on positive-biomarkers profile according to A/T/N classification [17]. Contrary to previous case–controls studies in which controls may have also included persons at a prodromal stage of AD, we chose a first control group drawn from two large European longitudinal cohorts: the Whitehall II study [19] and Three-City Study [20], in which prevalent and incident cases of dementia over the follow-up were excluded. Furthermore, the complementarity in terms of age of these two studies ensures a wide age range to match that of patients with AD. We also considered a second independent control group, recruited from individuals seen at the same memory clinics as the AD cases, who were characterized by a normal profile of cerebrospinal fluid (CSF) AD biomarkers.

## Methods

This study is reported following the Strengthening the Reporting of Observational Studies in Epidemiology (STROBE) guidelines (S1 STROBE Checklist). The study objectives and analysis plan were developed prior to data manipulation as an MSc internship project (S1 Text). The analyses undertaken in response to reviewers' suggestions are presented in post hoc analyses.

### Patients with AD

Data were drawn from nine memory centers in Europe (France [Paris, Rouen, Montpellier], Sweden [Gothenburg], Spain [Barcelona], Italia [Perugia], Belgium [Antwerp], and Germany [Göttingen and Munich]) and the ADNI study (see www.adni-info.org) [18]. Patients with AD included in the analyses were white and had a clinical diagnosis of probable AD [16], data on *APOE* genotype, and a positive-biomarkers profiles according to the A/T/N classification, and they are referred to as "CSF AD cases" in this study. Positivity of biomarkers profile was defined as abnormal values for CSF Aβ42 (A+) and CSF tau phosphorylated at threonine 181 (p-tau 181, T+), according to the reference values used in the memory centers. Assessment of CSF p-tau 181 was missing in one center (Gothenburg); biomarker positivity therefore relied only on CSF Aβ42 and CSF tau for this center. CSF Aβ40 was not measured, which precluded use of the amyloid ratio in the analyses. The number of patients included from each center as well as center-specific cutoffs of the CSF biomarkers are presented in S1 Table.

### Controls

"Population controls" came from two large European cohort studies of community-dwelling people with a focus on cognitive aging: the Whitehall II study (United Kingdom) and the Three-City Study (France). Design and procedures of these studies have been reported

previously [19,20]. Data on controls from these studies were drawn from the wave when *APOE* genotype was determined (baseline for Three-City Study, third clinical screening for Whitehall II). Both studies were complementary in terms of age of participants, as Whitehall II study included mainly middle-aged participants, and Three-City Study included persons 65 years or more at baseline. Therefore, controls less than 65 years old came from Whitehall II and those aged 65 years or more came from the Three-City Study. CSF biomarkers were not available in either of these studies. We excluded prevalent cases of dementia at measurement of *APOE* genotype and incident cases over the subsequent follow-up: 20 years in the Whitehall II study (168 incident cases) and 8 years in the Three-City Study (182 prevalent cases and 220 incident cases). In Whitehall II, dementia cases were ascertained based on linkage for all participants to electronic health records. Three registers (the national hospital episode statistics database, the Mental Health Services Data Set, and the mortality register) were used for dementia ascertainment by using ICD-10 codes F00-F03, F05.1, G30, and G31. In the Three-City Study, the detection of dementia cases was based on a three-step procedure at baseline and at each follow-up (neuropsychological evaluation, neurological examination, validation by an independent committee) as previously described [20]. Nonwhite participants were excluded in both studies.

We also used a second set of controls, identified as "CSF controls," consisting of white individuals recruited at the memory centers with normal values for all CSF biomarkers (Aβ42, tau, p-tau 181). This population consisted mainly of patients assessed for cognitive disorders other than AD and/or persons referred to the memory clinic who turned out not to have AD.

### Ethics statement

Ethical clearance was obtained by the institutional review boards of all participating sites (European memory centers and ADNI study sites). Ethical approval for the Whitehall II study was obtained from the NHS London—Harrow Research Ethics Committee (reference number 85/0938). The study protocol of Three-City Study was approved by the Ethical Committee of Kremlin-Bicetre University Hospital. All participants provided written, informed consent.

### CSF biomarkers assessment

For the European memory centres, CSF levels of Aβ42, total tau, and p-tau 181 were assessed with the commercially available sandwich ELISA INNOTEST, using the manufacturer's procedures (Fujirebio Europe NV, formerly Innogenetics NV).

In the ADNI study, CSF collection and processing are described in detail at http://adni.loni.usc.edu/methods/. Briefly, CSF Aβ42, tau, and p-tau 181 levels were quantified using multiplex xMAP Luminex platform (Luminex Corporation, Austin, TX) with Innogenetics (INNO-BIA AlzBio3; Ghent, Belgium) immunoassay kit–based research-use-only reagents containing 4D7A3 monoclonal antibody for Aβ42, AT120 monoclonal antibody for tau, and AT270 monoclonal antibody for p-tau 181.

The center-specific CSF biomarker cutoffs used in clinical setting were provided by each participating center and are presented in S1 Table.

### *APOE* genotyping

In the European memory centers, *APOE* genotype was determined using standard polymerase chain reaction and restriction enzyme digestion according to established standard protocols [21].

In the ADNI study, *APOE* genotypes were determined by using DNA extracted by Cogenics from a 3-mL aliquot of EDTA blood. Polymerase chain reaction amplification was followed by

HhaI restriction enzyme digestion, resolution on 4% Metaphor Gel, and visualization by ethidium bromide staining [22].

In Whitehall II and the Three-City Study, two TaqMan assays (Rs429358 and Rs7412, Assay-On-Demand, Applied Biosystems) were used and run on a 7900HT analyzer (Applied Biosystems), and *APOE* genotypes were indicated by the Sequence Detection Software version 2.0 (Applied Biosystems) [23,24].

## Covariates

Age in the analysis was set as age at CSF collection for AD cases and CSF controls and at blood collection for *APOE* genotyping in the population controls. As CSF biomarker assessment was performed as part of the AD diagnostic procedure, age at CSF collection was considered as a surrogate of age at diagnosis. Mini-Mental Status Examination (MMSE) score was taken from the date closest to the lumbar puncture for patients with AD and *APOE* determination in the population controls. Level of education was considered in three categories: no education to primary school, secondary school to high school, and baccalaureate or university degree.

In a subsample, data on the following cardiovascular risk factors were available: hypertension defined as systolic/diastolic blood pressure ≥140/90 mmHg or use of antihypertensive treatment; diabetes as fasting glycemia ≥1.26 g/L or use of antidiabetics drugs; and dyslipidemia as LDL cholesterol ≥1.9 g/L or use of lipid-lowering treatment.

## Statistical analyses

Participant characteristics were examined in three groups: CSF AD cases, population controls, and CSF controls. Proportions were calculated for categorical variables, and means and standard deviations were computed for continuous variables. Comparison of CSF AD cases with both control groups was assessed using a $\chi^2$ test or Student $t$ test as appropriate. Comparison of mean CSF biomarker values between CSF AD cases and CSF controls were adjusted for memory centers using analysis of covariance. Cumulative percentage of CSF AD cases as a function of age (in 5-year age groups) were plotted according to the *APOE* ε4 status (0, 1, 2 alleles); median age in the three groups was compared using the nonparametric Brown–Mood median test. We then used logistic regression to estimate the odds ratio (OR) of AD according to *APOE* genotype, comparing the total population of CSF-determined AD cases to population controls first and then to CSF controls. The role of age in modifying the association between *APOE* genotype and AD was examined in analyses stratified by 5-year age group, ranging from <55 years to >85 years. The ε3/ε3 group was the reference, and analysis was undertaken in groups using ε4 status. The nonlinearity of associations with AD as a function of age was tested using an interaction term between age$^2$ and APOE.

Population attributable fraction (PAF) associated with *APOE* ε4 (one allele, two alleles) for AD was plotted as a function of 5-year age group. PAF reflects the fraction of AD cases related to the presence of the *APOE* ε4 allele, and its calculation is based on prevalence of *APOE* ε4 among CSF AD cases ($PREV_{ε4}$) and the ORs associated with *APOE* ε4: PAF = $PREV_{ε4} \times (1 - 1/OR)$ [25]. ORs used for these estimations were those from the analysis comparing CSF AD cases to population controls.

We also examined whether sex and education affected the age-dependent association between *APOE* ε4 and AD using a triple interaction term in the logistic regression model. To ensure robustness of our findings, the analyses were repeated using controls defined by CSF biomarkers.

We conducted several post hoc analyses in response to peer review comments. First, as CSF p-tau 181 was missing for one of the centers (Gothenburg), we examined the impact of

excluding data from this center on our findings. Second, we calculated the sensitivity and specificity of carrying ≥1 APOE ε4 to discriminate CSF AD from controls, and the corresponding area under receiver operating characteristic (ROC) curves (AUCs) were plotted as a function of age to compare with previous studies [26]. Third, we estimated the heterogeneity of the association between *APOE* and AD in the various centers by stratifying the analyses by memory center and weighting their effect by the sample size, as in a meta-analysis. We calculated the $I^2$ statistic to evaluate heterogeneity in these estimates [27]. Finally, we examined the impact of cardiovascular risk factors on the association between *APOE* genotype and AD. These analyses were based on a subsample of the population of CSF AD cases for whom these data were available and all population controls. Using logistic regression, we first report crude ORs, followed by analyses adjusted for sex and education and, finally, for hypertension, diabetes mellitus, and hypercholesterolemia.

All resulting *p*-values were two-tailed, and $p \leq 0.05$ was considered statistically significant. Statistical analyses were performed using SAS version 9.4 (SAS Institute, Cary, NC, USA) and STATA version 14.

## Results

A total of 1,593 CSF AD cases, 11,723 population controls, and 805 CSF controls were included in the analyses; their characteristics are summarized in Table 1. Compared with CSF controls and population controls, CSF AD cases were older (72.8 versus 67.1 and 65.6 years), more

Table 1. Characteristics of the study population: CSF-determined AD cases, CSF-determined controls, and population controls.

| Characteristics | CSF AD cases (N = 1,593) | CSF controls (N = 805) | Population controls (N = 11,723) | *p*-Value[a] | *p*-Value[b] |
|---|---|---|---|---|---|
| Age, years, mean (SD) | 72.8 (8.3) | 67.1 (10.3) | 65.6 (10.7) | <0.001 | <0.001 |
| Women, *n* (%) | 883 (55.4) | 402 (49.9) | 5,525 (47.1) | 0.011 | <0.001 |
| MMSE, mean (SD)[c] | 21.4 (5.9) | 25.6 (4.4) | 27.5 (1.8) | <0.001 | <0.001 |
| Education, *n* (%)[d] | | | | 0.005 | <0.001 |
| Low | 83 (6.8) | 56 (9.3) | 612 (5.2) | | |
| Medium | 334 (27.3) | 195 (32.4) | 9,079 (77.4) | | |
| High | 805 (65.9) | 350 (58.2) | 2,032 (17.3) | | |
| *APOE* genotype, *n* (%) | | | | <0.001 | <0.001 |
| ε2/ε2 | 3 (0.2) | 2 (0.2) | 73 (0.6) | | |
| ε2/ε3 | 67 (4.2) | 113 (14.0) | 1,440 (12.3) | | |
| ε3/ε3 | 514 (32.3) | 518 (64.3) | 7,560 (64.5) | | |
| ε2/ε4 | 43 (2.7) | 8 (1.0) | 220 (1.9) | | |
| ε3/ε4 | 701 (44.0) | 161 (20.0) | 2,267 (19.3) | | |
| ε4/ε4 | 265 (16.6) | 3 (0.4) | 163 (1.4) | | |
| CSF biomarkers, pg/mL, mean (SD) | | | | | |
| CSF Aβ42 | 357.4 (177.5) | 803.6 (346.7) | — | <0.001[e] | |
| CSF tau | 535.3 (400.4) | 181.3 (88.7) | — | <0.001[e] | |
| CSF p-tau 181 | 83.2 (49.9) | 35.0 (12.8) | — | <0.001[e] | |

[a]Comparison between CSF AD cases and CSF controls, $\chi^2$ test or Student *t* test.

[b]Comparison between CSF AD cases and population controls, $\chi^2$ test or Student *t* test.

[c]MMSE data were missing for 108 CSF AD, 41 CSF controls, and 4,186 population controls.

[d]Education data were missing for 371 CSF AD and 204 CSF controls.

[e]Analyses on CSF biomarkers were adjusted for memory center and analysis of covariance.

Abbreviations: Aβ, β-amyloid; AD, Alzheimer disease; *APOE*, apolipoprotein E; CSF, cerebrospinal fluid; MMSE: Mini-Mental State Examination; p-Tau 181, tau phosphorylated at threonine 181; SD, standard deviation

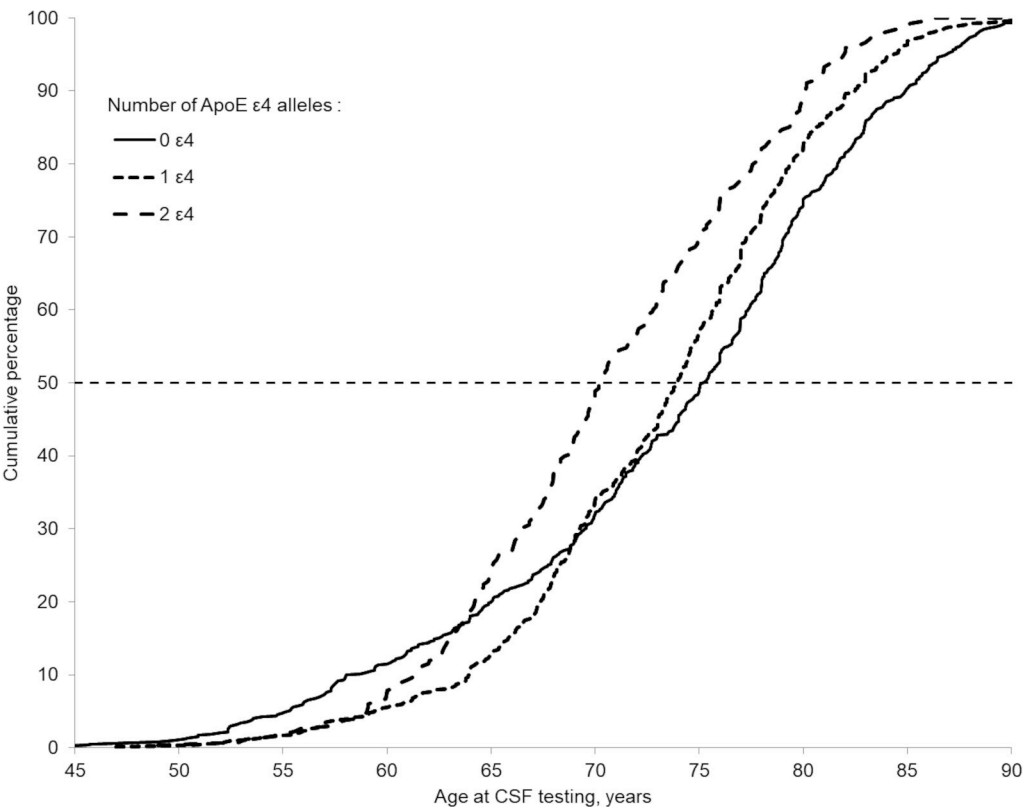

**Fig 1. Cumulative proportion of CSF AD cases as a function of age and *APOE* ε4 status.** The dashed horizontal line shows the median of distribution per group of *APOE* ε4 status. AD, Alzheimer disease; *APOE*, apolipoprotein E; CSF, cerebrospinal fluid.

likely to be women (55.4% versus 49.9% and 47.1%), and had a lower MMSE score (21.4 versus 25.6 and 27.5). CSF cases were also more likely to carry at least one *APOE* ε4 allele (46.7% versus 21.0% and 21.2%) or two ε4 alleles (16.6% versus 0.4% and 1.4%). The proportion of *APOE* ε4 carriers did not differ between population controls and CSF controls (respectively, 22.6% and 21.4%, $p = 0.41$).

Fig 1 presents the cumulative proportion of CSF AD cases as a function of age and *APOE* ε4 status. The median age at AD diagnosis of non-ε4 carriers (75.2 years) was greater than that of one ε4 carriers (73.8 years, $p = 0.016$) and of ε4/ε4 carriers (70.3 years, $p < 0.001$). Compared with non-ε4 carriers, the cumulative percentage of AD cases in two ε4 carriers was higher starting at age 63.5 years, and in one ε4 carriers, the percentage was higher starting at 69.1 years.

The unadjusted association of *APOE* genotypes with AD is presented in Table 2. The presence of at least one ε4 allele was associated with an OR of 5.9 (95% confidence interval [CI] 5.3–6.6; $p < 0.001$) compared with non-ε4 carriers. Using ε3/ε3 as the reference, ε2 allele was associated with lower risk of AD (OR: 0.68 [0.53–0.88]; $p = 0.003$), whereas ε2/ε4 (OR: 2.9 [2.0–4.0]; $p < 0.001$), ε3/ε4 (OR: 4.5 [4.0–5.1]; $p < 0.001$), and ε4/ε4 (OR: 23.9 [19.3–29.6]; $p < 0.001$) were associated with greater risk of AD. Using CSF controls instead of population controls yielded similar findings (Table 2), as well as the exclusion of the center with missing data for CSF phosphorylated tau (see S2 Table).

The proportion of *APOE* ε4 carriers by 5-year age group is shown in Fig 2. In total, 28.9% of CSF AD cases <55 years at diagnosis carried one ε4 allele, and this rate increased

**Table 2. The ORs of AD according to *APOE* genotype.** CSF AD cases were compared with population controls and with CSF controls using logistic regression analysis.

| *APOE* genotype | CSF AD cases versus population controls | | CSF AD cases versus CSF controls | |
|---|---|---|---|---|
| | OR (95% CI) | *p*-Value | OR (95% CI)[a] | *p*-Value[a] |
| 0 ε4 | 1 (Ref) | | 1 (Ref) | |
| ≥1 ε4 | 5.9 (5.3–6.6) | <0.001 | 6.4 (5.2–7.7) | <0.001 |
| 0 ε4 | 1 (Ref) | | 1 (Ref) | |
| 1 ε4 | 4.6 (4.1–5.2) | <0.001 | 4.8 (3.9–5.8) | <0.001 |
| 2 ε4 | 25.4 (20.4–31.2) | <0.001 | — | — |
| ε2/ε2, ε2/ε3 | 0.68 (0.53–0.88) | 0.003 | 0.61 (0.44–0.85) | 0.003 |
| ε3/ε3 | 1 (Ref) | | 1 (Ref) | |
| ε2/ε4 | 2.9 (2.0–4.0) | <0.001 | — | — |
| ε3/ε4 | 4.5 (4.0–5.1) | <0.001 | 4.4 (3.6–5.4) | <0.001 |
| ε4/ε4 | 23.9 (19.3–29.6) | <0.001 | — | — |

OR could not be determined due to small numbers in the following CSF control categories: ε4/ε4 (*n* = 3) and ε2/ε4 (*n* = 8) categories.

Abbreviations: AD, Alzheimer disease; *APOE*, apolipoprotein E; CI, confidence interval; CSF, cerebrospinal fluid; OR, odds ratio; Ref, reference

progressively to 54.2% in the group aged 65–70 years and then progressively decreased to 28.1% in those over 85 years at AD diagnosis. Similarly, 8.9% of AD cases <55 years at diagnosis were homozygous ε4/ε4; this proportion climbed progressively to 29.3% in the group aged 60–65 years and then progressively decreased (3.7% in those ≥85 years). Among population controls, the proportion of *APOE* ε4 carriers was around 25% until 65 years old, 20% between 65 and 80 years old, and 15% after the age of 80 years. The distribution of *APOE* ε4 as a function of age was similar in population controls and CSF controls ($\chi^2$ test, *p* = 0.42).

The PAF associated with the presence of at least one *APOE* ε4 allele for AD in the total population was 53%; the PAF by 5-year age categories is presented in Fig 3. PAF associated with at least one ε4 was highest in the group aged 65–70 years, at 69.7%. The PAF associated with one ε4 allele was also highest in the group aged 65–70 years, at 48.0%. The PAF of two ε4 alleles was highest in the group aged 60–65 years, at 28.4%. The importance of ε4 for AD was lower both at younger (<55 years, PAF: 14.2%) and older (>85 years, PAF: 22.6%) ages.

Fig 4 shows the association of *APOE* genotype with AD as a function of 5-year age group, adjusted for sex and education. There was no evidence that the age-related association of APOE ε4 differed by sex (*p* = 0.77) or education (*p* = 0.61). The association of APOE ε4 genotype with AD was modified by age (*p* for interaction with $age^2$ < 0.001), with the strongest association in the 65–70 age group: one ε4 allele (OR: 10.7 [7.6–15.2]; *p* < 0.001), ε4/ε4 (OR: 64.2 [34.1–120.8]; *p* < 0.001). The protective association of *APOE* ε2 genotype for risk of AD was not affected by age (*p* for interaction = 0.72).

## Post hoc analysis

Sensitivity and specificity of *APOE* ε4 to discriminate CSF AD from population controls was, respectively, 56% and 78% for heterozygotes and 31% and 98% for homozygotes, corresponding to an AUC of 0.70 (0.69–0.72, *p* < 0.001); similar results were found when comparing CSF AD cases with CSF controls (AUC = 0.70 [0.69–0.73], *p* < 0.001). The ability of *APOE* ε4 to discriminate AD from controls differed as a function of age (Fig 5) and was greatest in the

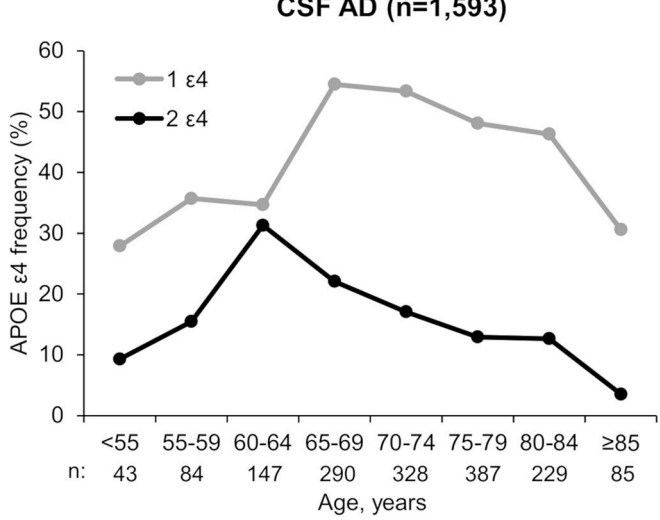

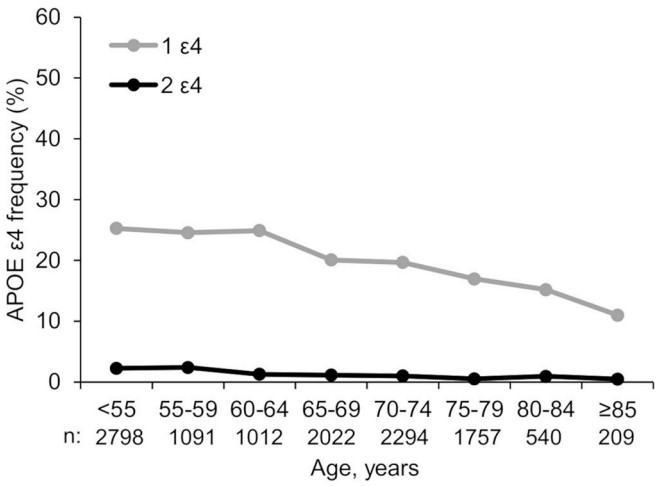

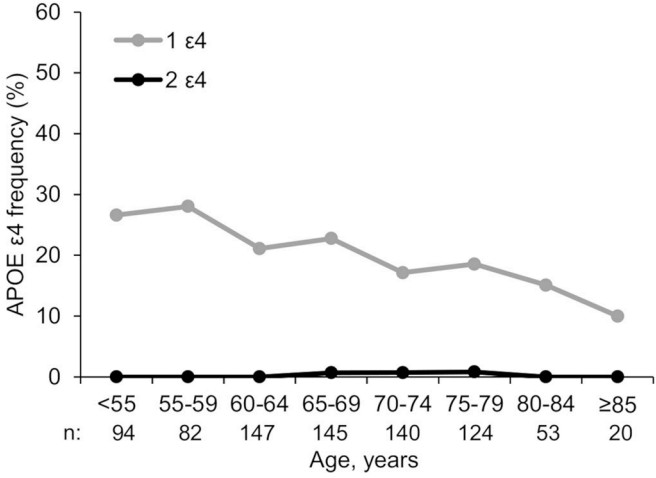

**Fig 2. *APOE* ε4 prevalence in CSF AD cases and population and CSF controls as a function of age group.** AD, Alzheimer disease; *APOE*, apolipoprotein E; CSF, cerebrospinal fluid.

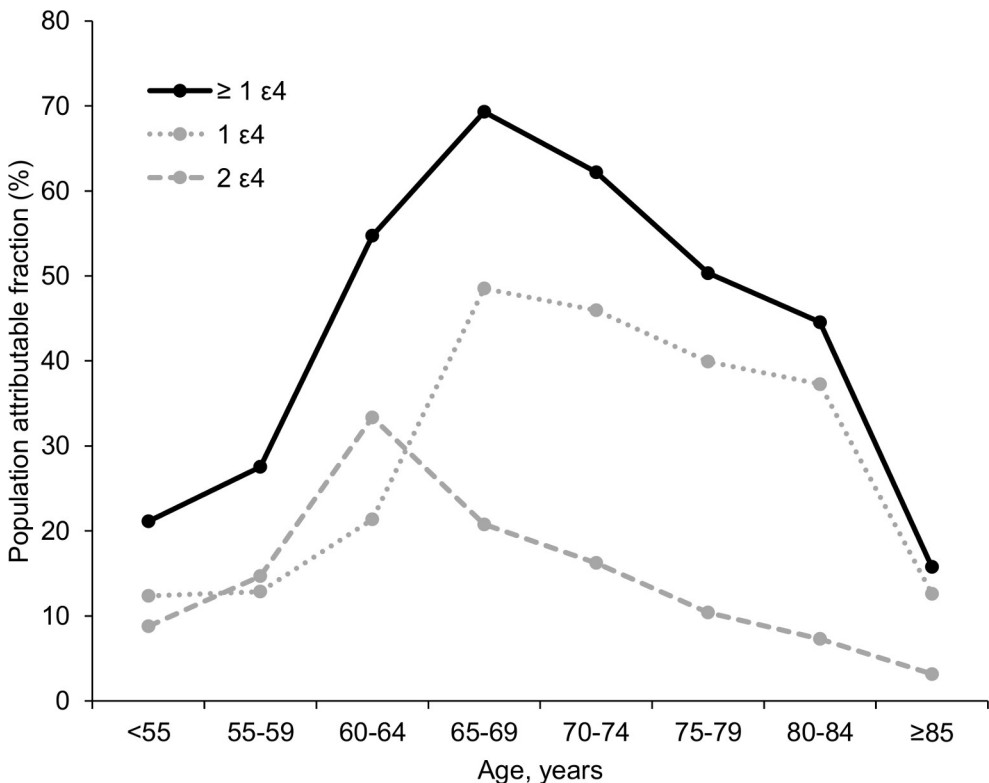

**Fig 3. Population attributable fraction of *APOE* ε4 for AD by 5-year age group.** AD, Alzheimer disease; *APOE*, apolipoprotein E.

group aged 65–69 years (AUC = 0.78 [0.75–0.80], $p < 0.001$) and poorer before 55 years (AUC = 0.55 [0.47–0.62], $p = 0.32$) and after 85 years (AUC = 0.61 [0.56–0.67], $p = 0.001$).

We examined the heterogeneity in associations of one or more *APOE* ε4 with AD across the different memory centers using population controls as the comparison group. These results are shown in Fig 6. The $I^2$ statistic was 58%, corresponding to some heterogeneity between centers.

The final analysis was on the subset of patients with AD (841 CSF AD cases) with data on cardiovascular risk factors. There were no significant differences between CSF AD cases with and without these data in term of age ($p = 0.80$) and *APOE* genotype ($p = 0.67$). Compared with general population controls (see S3 Table), CSF AD cases were more likely to have hypertension ($p = 0.024$), diabetes mellitus ($p < 0.001$), and hypercholesterolemia ($p < 0.001$). We examined the role of these risk factors in the association of *APOE* genotypes with AD without adjustment for age due to its role in modifying the association between *APOE* genotype and AD. Table 3 shows these results. Adjustment for sex and education (model 1) and for cardiovascular risk factors (model 2) had little impact on the estimates compared with the unadjusted model. S1 Fig shows similar findings in analyses stratified by age group.

## Discussion

In this large multicentric study that included 1,600 biomarker-positive AD cases, we showed that the association between *APOE* ε4 and AD is modified by age. The impact of ε4 was less pronounced before the age of 60, strongest between 65 and 70 years, and then declined progressively at older ages. *APOE* ε4 carriers were more likely to develop AD at younger ages,

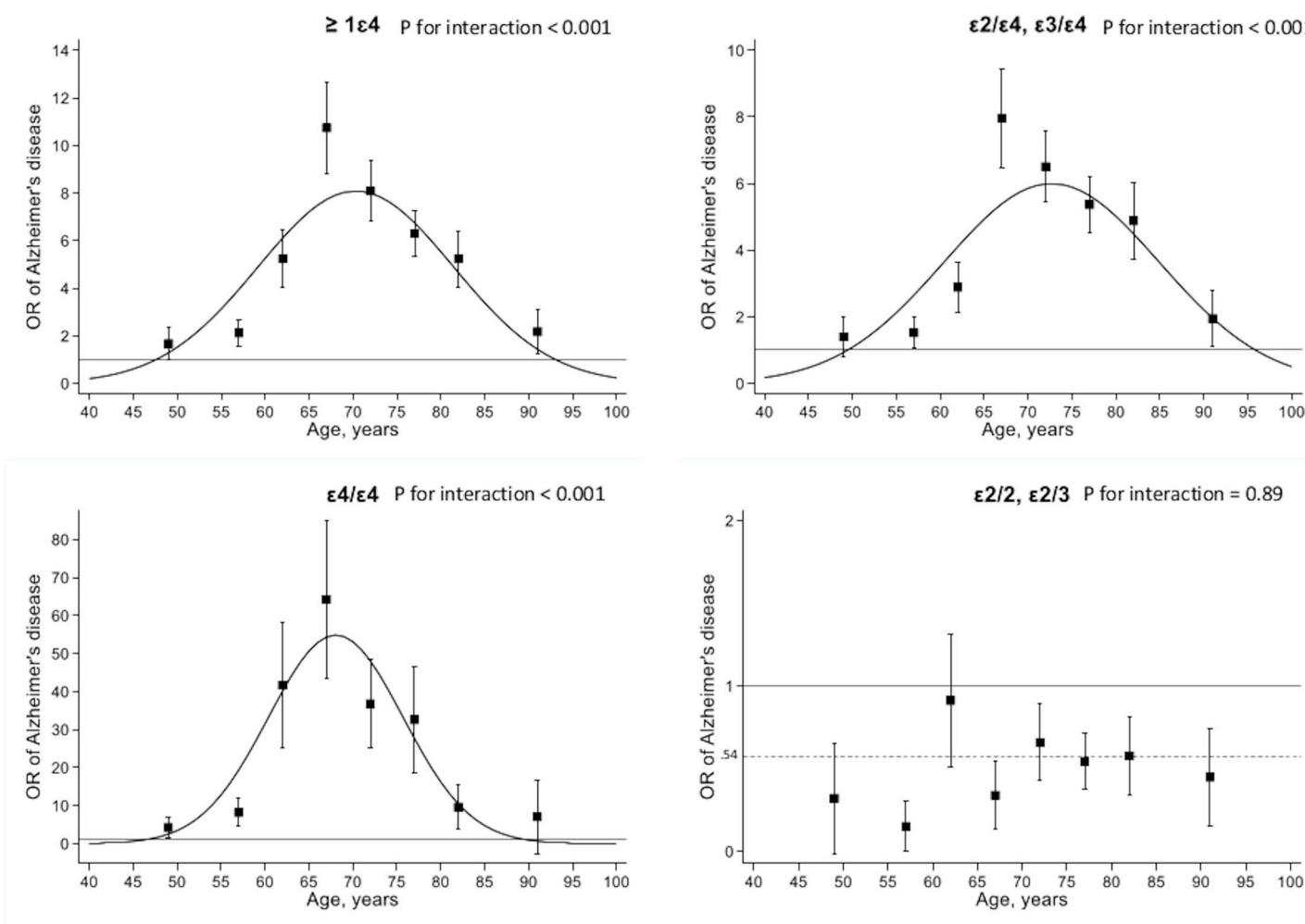

**Fig 4. Association of *APOE* genotype in CSF-determined AD cases compared with population controls.** Genotype ε3/ε3 was used as reference. *p*-Value is for interaction between age$^2$ and *APOE*. Error bars correspond to the standard errors of the ORs calculated by 5-year age categories. Association between age and OR of AD (curves in the graphs) was modeled using a quadratic term for age in the logistic regression model and adjusted for sex and education. AD, Alzheimer disease; *APOE*, apolipoprotein E; CSF, cerebrospinal fluid; OR, odds ratio.

with a difference of 4.9 years for ε4/ε4 and 1.4 years for one ε4 carriers when compared with median age at AD diagnosis in non-ε4 carriers (75.2 years). The PAF associated with *APOE* ε4 for AD was 53% overall, but it varied strongly with age, following a bell-curve relationship that ranged from less than 20% in the youngest and the oldest age groups and reaching 70% between 65 and 70 years. The protective effect of *APOE* ε2 allele was unaffected by age. The association between *APOE* genotype and AD was first reported in 1993 [28], and age is thought to play a role in this association [29,30]. Our use of a large, multicentric study of patients with AD defined by the new A/T/N criteria allows the risk of misclassification bias to be addressed, as it can be critical in such analyses. A further advantage of our study is the use of population controls from two large longitudinal cohorts in which the follow-up allowed us to remove incident cases of dementia. Because this control group without clinical dementia may be positive on AD biomarkers, we also considered a second group of controls drawn from the same memory clinics as the AD cases. This group of controls had a normal CSF AD biomarker profile,

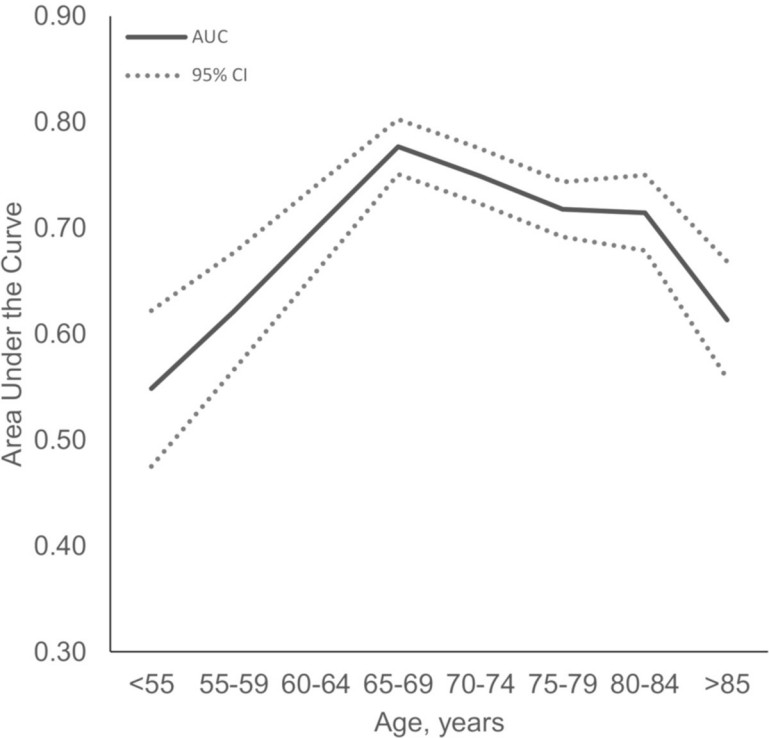

**Fig 5. Analysis of the AUC in function of age for at least one *APOE* ε4 carrying versus none to discriminate CSF AD from population controls.** AD, Alzheimer disease; *APOE*, apolipoprotein E; AUC, area under receiver operating characteristic curve; CI, confidence interval; CSF, cerebrospinal fluid.

and results using these two independent sets of controls were similar, suggesting that our findings are robust.

In our study, carrying at least one APOE ε4 allele was associated with a 5.9 (5.3–6.6; $p < 0.001$) higher OR of AD, compared with an OR of 3.3 (3.2–3.4; $p < 0.001$) reported by a recent meta-analysis of 46 case–control studies with 64,000 participants [31]. We found the overall PAF associated with one or more *APOE* ε4 allele to be 53%, compared with the 37% reported by the AlzGene meta-analysis [32]. Our estimates are higher than those previously reported from studies based on only clinical AD criteria [33] but are close to those reported when using neuropathology [34] or biomarker approaches for AD diagnoses [35,36]. Our results, along with recent findings from other studies, suggest that the impact of *APOE* ε4, particularly ε4 homozygotes, is underestimated in studies based exclusively on clinical criteria for AD diagnosis [37]. A possible explanation is the importance of *APOE* ε4 specifically for AD neuropathology [38]. Another explanation is linked to the key finding that the association between *APOE* ε4 and AD is mediated by age. Therefore, a single, overall OR may not be suitable; reporting age-specific ORs, as in our analyses, may facilitate comparison between studies.

Our findings have implications for several types of studies, e.g., epidemiological and genetic, aiming to reveal biological associations between environmental and genetic risk factors for AD, which generally are performed on clinically diagnosed patients and cognitively unimpaired controls, without biomarker data. Midlife vascular risk factors (obesity, smoking, diabetes, hypertension, and cardiac disease) have commonly been found to increase risk of cognitive decline and AD dementia defined using clinical criteria; population studies employing biomarkers show that such risk factors are associated with neurodegeneration but not brain amyloidosis [39]. Further, genome-wide association studies have identified loci

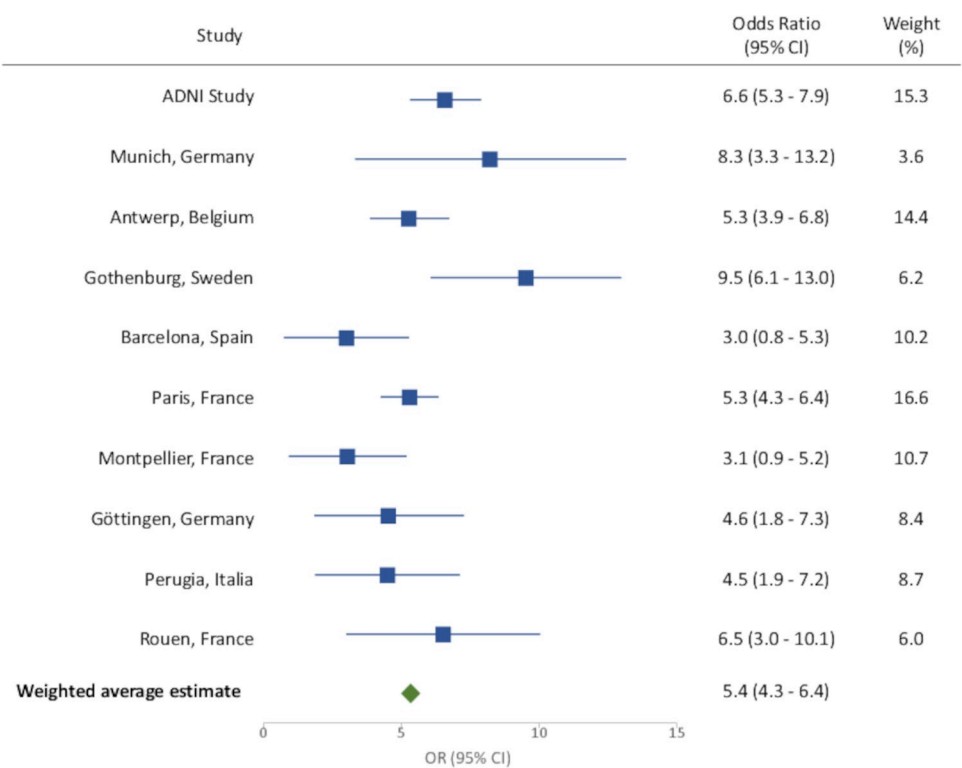

**Fig 6. Association between *APOE* ε4 carrying and AD stratified by center, using general population group as controls.** AD, Alzheimer disease; *APOE*, apolipoprotein E; CI, confidence interval; OR, odds ratio.

associated with clinically defined AD, but when examined in autopsy cohorts, no associations were found with plaques or tangle pathology; instead, associations were found with

**Table 3. The ORs of AD in CSF AD cases compared with population controls: multivariable logistic regression analysis.**

| *APOE* | Unadjusted | | Model 1[a] | | Model 2[b] | |
|---|---|---|---|---|---|---|
| | OR (95% CI) | *p*-Value | OR (95% CI) | *p*-Value | OR (95% CI) | *p*-Value |
| 0 ε4 | 1 (Ref) | | 1 (Ref) | | 1 (Ref) | |
| ≥1 ε4 | 5.9 (5.1–6.8) | <0.001 | 6.0 (5.1–7.0) | <0.001 | 6.0 (5.1–7.0) | <0.001 |
| 0 ε4 | 1 (Ref) | | 1 (Ref) | | 1 (Ref) | |
| 1 ε4 | 4.7 (4.0–5.4) | <0.001 | 4.7 (4.0–5.6) | <0.001 | 4.7 (4.0–5.6) | <0.001 |
| 2 ε4 | 24.6 (19.0–31.7) | <0.001 | 25.7 (19.0–34.7) | <0.001 | 26.5 (19.5–36.1) | <0.001 |
| ε2/ε2, ε2/ε3 | 0.55 (0.38–0.81) | 0.002 | 0.54 (0.37–0.79) | 0.002 | 0.53 (0.36–0.78) | 0.001 |
| ε3/ε3 | 1 (Ref) | | 1 (Ref) | | 1 (Ref) | |
| ε2/ε4 | 2.5 (1.5–3.9) | <0.001 | 2.3 (1.4–3.9) | <0.001 | 2.3 (1.4–3.8) | 0.001 |
| ε3/ε4 | 4.5 (3.8–5.3) | <0.001 | 4.6 (3.8–5.4) | <0.001 | 4.6 (3.8–5.5) | <0.001 |
| ε4/ε4 | 22.7 (17.6–29.4) | <0.001 | 23.7 (17.4–32.1) | <0.001 | 24.5 (18.0–33.4) | <0.001 |

Analyses were performed on a subsample (*N* = 841 AD cases, 11,665 controls).

[a]Model 1: adjustment for sex and education.

[b]Model 2: model 1 + further adjustment for hypertension, diabetes, and hypercholesterolemia.

Abbreviations: AD, Alzheimer disease; APOE, apolipoprotein E; CI, confidence interval; CSF, cerebrospinal fluid; OR, odds ratio; Ref, reference

cerebrovascular disease [40]. Thus, there is a need to reinvestigate possible associations between environmental and genetic risk factors for AD using biomarker-based studies to understand the biologic basis of such associations.

Causes of early-onset AD, before 65 years, remain poorly understood. Autosomal dominant mutations, affecting amyloid precursor protein (*APP*), presenilin-1 (*PSEN1*), or presenilin-2 (*PSEN2*) genes, are thought to be involved, but these mutations are rare, accounting for less than 10% of early-onset AD [1]. During this past decade, several genome-wide association studies have discovered around 30 risk loci of the disease, which are useful to identify novel insights into the neurobiology of the disease, but their contribution to explaining AD in the population is thought not to be substantial [10]. Conversely, recent efforts in whole-exome and whole-genome sequencing of large AD case–control series unraveled three major genes— sortilin-related receptor (*SORL1*); triggering receptor expressed on myeloid cells-2 (*TREM2*); and ATP-binding cassette, sub-family A, member 7 (*ABCA7*)—the rare coding variants of which significantly increase the risk of AD with moderate to high OR [41]. Interestingly, the ORs were higher in patients with early-onset AD [41], and some patients carried more than one strong risk factor, including *APOE* ε4, suggesting an oligogenic inheritance in some of these patients.

In our study, *APOE* ε4 did not play a strong role in risk of AD before the age of 60 years, whereas its role increased dramatically during the seventh decade to hit its peak by 70 years. Asymptomatic *APOE* ε4 carriers are a major target for potential disease-modifying drugs to prevent AD, and several clinical trials are currently ongoing on this population [42]. Our findings support the idea of an early intervention in this population, if and when available, before *APOE* ε4 carriers reach the age at which incidence of AD rises sharply.

## Strengths and limitations

This study has several strengths, including the use of AD cases determined using CSF biomarkers for tau and Aβ; a large sample size, which allowed sufficient power to undertake analysis in 5-year age categories; as well as the use of two large population cohort studies with long follow-up data, allowing us to exclude incident cases of dementia. Furthermore, we were able to replicate findings using a further set of controls, consisting of persons with normal CSF biomarkers who were seen at the same memory centers as the AD cases. The distribution of *APOE* genotypes by age group was similar in the two sets of controls, and the associations with AD using the two types of controls were also similar.

This study needs to be considered in light of the following limitations. First, AD cases drawn from memory centers may not be representative of the general population of patients with AD, in particular patients who do not seek medical care. This may involve some selection bias, primarily due to sociodemographic factors like education or sex, but it is unlikely to be a source of major bias because *APOE* status is unlikely to affect the decision to consult at memory clinics. Second, data on cardiovascular risk factors or familial history were missing for a part of the CSF AD cases, and adjustment for these risk factors was possible only on a subsample. More-detailed data on a larger set of risk factors would be useful in future studies to better understand the mechanisms underlying the relationship between *APOE* and AD. Third, the overall ratio between AD cases and controls used in this study was around 1:7, which may lead to detection of nonpertinent associations due to an overpowered design. However, the value of a large sample size is the ability to examine the association between *APOE* and AD in age subgroups to show the manner in which age modifies this association [43]. Finally, these analyses were restricted to white populations because the *APOE* ε4 allele frequency as well as its association with AD risk may be strongly influenced by ethnicity and geographic region [44,45].

Further research using different population settings is needed to better understand these complex associations.

## Conclusions

This study supports an association between *APOE* ε4 and the risk of AD evaluated using biomarkers of AD neuropathology, and the results suggest that the strength of this association varies by age. The effect of *APOE* ε4 on AD was less pronounced in both younger and older persons (less than 60 and more than 85 years) and was greatest between 65 and 70 years, with a population attributable risk of 70% in this group. These findings highlight the importance of early interventions, if and when available, to mitigate the effect of *APOE* ε4 on AD.

## Supporting information

**S1 STROBE Checklist. STROBE checklist for cohort studies.** STROBE, Strengthening the Reporting of Observational Studies in Epidemiology.
(DOCX)

**S1 Fig. Association of *APOE* genotype in CSF AD cases compared with population controls in a subsample with data on cardiovascular risk factors.** Genotype ε3/ε3 was used as reference. Associations between age and OR of AD were modeled using a quadratic term for age in the logistic regression model and adjusted for sex, education, hypertension, diabetes mellitus, and hypercholesterolemia. AD, Alzheimer disease; *APOE*, apolipoprotein E; CSF, cerebrospinal fluid; OR, odds ratio.
(TIF)

**S1 Table. Threshold levels used for defining abnormal values of CSF Aβ42, CSF tau, and CSF p-tau 181 in the memory centers that participated in the study.** Aβ, β-amyloid; CSF, cerebrospinal fluid; p-Tau 181, tau phosphorylated at threonine 181.
(DOCX)

**S2 Table. The odds ratios of AD in CSF AD cases compared with population controls before and after exclusion of the center with missing CSF phosphorylated tau.** AD, Alzheimer disease; CSF, cerebrospinal fluid.
(DOCX)

**S3 Table. Characteristics of the subsample used for the assessment of the association between *APOE* genotype and AD adjusted for cardiovascular risk factors.** AD, Alzheimer disease; *APOE*, apolipoprotein E.
(DOCX)

**S1 Text. Prospective analysis plan.**
(DOCX)

## Acknowledgments

Some of the data used in preparation of this article were obtained from the ADNI database (adni.loni.usc.edu). As such, the investigators within the ADNI contributed to the design and implementation of ADNI and/or provided data but did not participate in analysis or writing of this report. A complete listing of ADNI investigators can be found at http://adni.loni.usc.edu/wp-content/uploads/how_to_apply/ADNI_Acknowledgement_List.pdf. ADNI data are disseminated by the Laboratory for Neuro Imaging at the University of Southern California. The Three-City Study is conducted under a partnership agreement between the Institut National

de la Santé et de la Recherche Médicale (INSERM), the Victor Segalen-Bordeaux II University, and the Sanofi-Synthélabo Company.

## Author Contributions

**Conceptualization:** Hana Saddiki, Aurore Fayosse, Emmanuel Cognat, Julien Dumurgier.

**Formal analysis:** Aline Dugravot, Julien Dumurgier.

**Funding acquisition:** Aurore Fayosse.

**Investigation:** Emmanuel Cognat, Panagiotis Alexopoulos, Kaj Blennow, Henrik Zetterberg, Lucilla Parnetti, Inga Zerr, Peter Hermann, Audrey Gabelle, Mercè Boada, Adelina Orellana, Itziar de Rojas, Matthieu Lilamand, Maria Bjerke, Christine Van Broeckhoven, Lucia Farotti, Nicola Salvadori, Janine Diehl-Schmid, Timo Grimmer, Claire Hourregue, Aline Dugravot, Gaël Nicolas, Jean-Louis Laplanche, Sylvain Lehmann, Elodie Bouaziz-Amar, Jacques Hugon, Christophe Tzourio, Archana Singh-Manoux, Claire Paquet, Julien Dumurgier.

**Methodology:** Aurore Fayosse, Séverine Sabia, Sebastiaan Engelborghs, Archana Singh-Manoux, Julien Dumurgier.

**Resources:** Séverine Sabia, Sebastiaan Engelborghs, David Wallon, Panagiotis Alexopoulos, Kaj Blennow, Henrik Zetterberg, Lucilla Parnetti, Inga Zerr, Peter Hermann, Audrey Gabelle, Mercè Boada, Adelina Orellana, Itziar de Rojas, Matthieu Lilamand, Maria Bjerke, Christine Van Broeckhoven, Lucia Farotti, Nicola Salvadori, Janine Diehl-Schmid, Timo Grimmer, Claire Hourregue, Aline Dugravot, Gaël Nicolas, Jean-Louis Laplanche, Sylvain Lehmann, Elodie Bouaziz-Amar, Jacques Hugon, Christophe Tzourio, Archana Singh-Manoux, Claire Paquet, Julien Dumurgier.

**Supervision:** Hana Saddiki, Archana Singh-Manoux, Julien Dumurgier.

**Validation:** Emmanuel Cognat, Sebastiaan Engelborghs, David Wallon, Panagiotis Alexopoulos, Kaj Blennow, Lucilla Parnetti, Julien Dumurgier.

**Visualization:** Julien Dumurgier.

**Writing – original draft:** Hana Saddiki, Aurore Fayosse, Emmanuel Cognat, Séverine Sabia, Kaj Blennow, Henrik Zetterberg, Jacques Hugon, Christophe Tzourio, Archana Singh-Manoux, Claire Paquet, Julien Dumurgier.

**Writing – review & editing:** Hana Saddiki, Aurore Fayosse, Emmanuel Cognat, Séverine Sabia, Sebastiaan Engelborghs, David Wallon, Panagiotis Alexopoulos, Kaj Blennow, Henrik Zetterberg, Lucilla Parnetti, Inga Zerr, Peter Hermann, Audrey Gabelle, Mercè Boada, Adelina Orellana, Itziar de Rojas, Matthieu Lilamand, Maria Bjerke, Christine Van Broeckhoven, Lucia Farotti, Nicola Salvadori, Janine Diehl-Schmid, Timo Grimmer, Claire Hourregue, Aline Dugravot, Gaël Nicolas, Jean-Louis Laplanche, Sylvain Lehmann, Elodie Bouaziz-Amar, Jacques Hugon, Christophe Tzourio, Archana Singh-Manoux, Claire Paquet, Julien Dumurgier.

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
