## [Decision Letter · Decision Letter 0]

17 Feb 2020

Dear Dr. Dumurgier,

Thank you very much for submitting your manuscript "Age modifies the association between Apolipoprotein E genotype and Alzheimer’s disease: a multicountry cerebrospinal fluid biomarker-based case-control study." (PMEDICINE-D-19-04199) for consideration at PLOS Medicine. 

Your paper was discussed among the editorial team, evaluated by an academic editor with relevant expertise, and sent to independent reviewers, including a statistical reviewer. The reviews are appended at the bottom of this email and any accompanying reviewer attachments can be seen via the link below:

[LINK]

In light of these reviews, we will not be able to accept the manuscript for publication in the journal in its current form, but we would like to invite you to submit a revised version that fully addresses the reviewers' and editors' comments. You will appreciate that we cannot make a decision about publication until we have seen the revised manuscript and your response, and we expect to seek re-review by one or more of the reviewers. 

We hope to receive your revised manuscript by Mar 09 2020 11:59PM. Please email us (plosmedicine@plos.org) if you have any questions or concerns.

Please let me know if you have any questions. Otherwise, we look forward to receiving your revised manuscript in due course. 

Sincerely,

Richard Turner PhD, for Louise Gaynor-Brook, MBBS PhD

Associate Editor, PLOS Medicine

rturner@plos.org

Please adapt the title so that it is not declarative. We suggest "Age and the association between apolipoprotein E genotype and Alzheimer’s disease: a cerebrospinal fluid biomarker-based case-control study.

In the abstract and elsewhere, please add p values alongside CI where available.

Please add summary demographic details for study participants to the abstract.

Please add a new final sentence to the "methods and findings" subsection of your abstract, which should summarize the study's main limitations. 

Please begin the sentence at line 86 with "In this study, we found that ..." or similar. 

After the abstract, we will need to ask you to add a new and accessible "author summary" section in non-identical prose. You may find it helpful to consult one or two recent research papers published in PLOS Medicine to get a sense of the preferred style. 

Early in the methods section of your main text, please state whether the present study had a protocol or prespecified analysis plan, and if so attach the relevant document(s) as an attachment, referred to in the text. please highlight analyses that were not prespecified. 

At line 345 and any other instances, please avoid claims of "the first" or similar, and where these are needed please add "to our knowledge", for example. 

Where study limitations are summarized (at line 412), we suggest crafting a dedicated paragraph and providing a fuller discussion. 

In the reference list, please adapt citations so that, where appropriate, six rather than five authors' names are listed, followed by "et al.".

For reference 21, please make the journal name "BMJ".

Please adapt the attached STROBE checklist so that individual items are referred to by section (e.g., "Methods") and paragraph number rather than by page or line numbers, as the latter generally change in the event of publication. Please refer to this in the methods section of your main text. 

Comments from the reviewers:

*** Reviewer #1: 

Thank you for the opportunity to review this very interesting and compelling study by Saddikki and colleagues on exploring the association between the APOE e genotype and dementia. This retrospective study used brought together a consortia of AD patients and non-AD controls from Whitehall II 3-City to determine the effect size of association between APOE e4 allele (one/two copies) and AD, provide PAFs by age-stratum. The author did find an significant association (with stronger effects seen in the homozygotes) with the effect modified by age. The effect sizes were much higher than previously reported in published evidence, however it has been well-established from previous meta-analysis and case-control studies that APOE genotypes are associated strongly with AD. The case control design is an appropriate design for this analysis and the statistical analysis was simple and straightforward. 

However, my major comments regards on controls selection, imbalance between cases and controls which will require some form of covariate adjustment or propensity score matching as the effect sizes of unadjusted ORs may be overestimated.

Specific comments related to this:

1) The authors utilised two large community dwelling cohorts to source controls from in the UK and France with APOE genotyping with 11,724 controls. The ratio between AD cases and controls is therefore approx. 1:7. Previous evidence has suggested that ORs for APOE genotypes are order of magnitude of > 3. I'm surprised the authors did not report a sample size calculation based on expected detection to get a sense of the minimum ratio of controls to cases. I mention this because it's likely the study is overpowered. While larger number of controls may seem obviously better, there may have been an opportunity to do a more sophisticated selection technique to account for selection and indication bias (which the authors themselves discuss as a limitation) given set a large number of controls. Authors clarification and comment on this aspect would be useful here. 

2) Related the above comment then, I bring up the control selection because in Table 1, there are significant imbalances in age, gender, MMSE, education, CSF biomarkers (P < 0.001) and these are just the covariates that were reported. I would venture to guess that there are quite few more covariates captured. In the subsequent analysis of the logistic regression models (Table 2), there was no indication any covariate adjustments were made. In fact, there seems to be no consideration of prior history vascular conditions (for instance APOE genotypes are associated with familial hypercholesterolaemia and atherosclerosis - https://www.ncbi.nlm.nih.gov/pmc/articles/PMC5505295/). Reporting crude ORs is good but what would improve the analysis and be more convincing if adjusted analyses could also be provided to account for confounders. 

3) One method that could help deal with residual confounding is a the use of propensity scores adjustment (https://www.sciencedirect.com/science/article/pii/S1476927116300135) - creating a regression model (based on distribution of all the covariates) and then using the score to adjust during the model. This does not necessarily need to be done if simple covariate adjustment, considering confounding factors such as vascular comorbidities was taken into account. Barring that, covariate adjustment in adjusted model should be considered. 

4) An early study with similar design (https://www.sciencedirect.com/science/article/pii/0895435696001953) showed similar results but looked at the sensitivity and specificity of E4 genotypes and found that the sensitivity was 0.52 /specificity was 0.74 for heterozygotes and homozygotes 0.23 sensitivity/0.99 specificity. The authors concluded that it was significant risk factors but as a diagnostic marker for AD would miss many true cases and misclassify many normal as AD. Can the authors show what the sensitivity and specificity of the biomarker is in their sample and whether or not they believe it has additional added value over and beyond the clinical assessment. Ideally you want to perform a ROC analyses but understand this may be not in the scope of the study analysis. 

Apart from these issues above, the study reveals that the effect sizes were similar but perhaps there conclusion from their crude ORs that the associations between APOE genotypes and AD is much greater than previously reported may actually decrease when consideration of better control selection and covariate adjustment is taken into account. It would be advisable for the authors to consider these issues prior to consideration for publication. 

*** Reviewer #2: 

In this work Saddiki and colleagues aimed to assess the association between AD and APOE genotype in groups of patients based on age-specific increments. In this confirmatory study the studied subjects were included from 9 different European centers, and the ADNI. The rationale for the study is based on previous studies assessing associations between APOE genotype and AD in patients diagnosed based on clinical criteria rather than evidence of AD pathology as assessed by available biomarkers - hence the authors speculated that the previously reported strong association between APOE and AD is an underestimation due to imprecise clinical diagnosis. Results are in line with previous studies reporting a strong association between APOE4 and AD and which is modified by age - less pronounced effect in younger and older individuals. The main take-home is that the association between AD and a homozygous APOE4 genotype is stronger than that reported in previous studies which were based on clinical AD diagnosis without the use of CSF biomarkers. This is an interesting study which is largely confirming previous studies hence novelty is limited yet the take-home message is of importance. This reviewer has several major concerns which need to be addressed before this work can be accepted for publication:

Major:

- The studied subjects were assessed in various different memory clinics which according to the supplementary information used locally-defined CSF biomarker cut-offs which at least for Abeta varied dramatically. Given the variation in AD risk between subjects in different geographical locations (most probably due to impact of environment factors like diet, sun exposure etc) the authors are asked to include a table specifying the number of subjects included from each European center so that the effect of each location on the results can be assessed. This is important especially since the results may be biased by results from specific populations in which amyloid-beta pathology (which is driven mainly by presence of the APOE4 allele both in patients and controls) was more/less pronounced as assessed by CSF AD biomarkers.

- Subjects from one of the centers lacked CSF p-tau and amyloid-beta40, the authors need to explain how this affected the A/T/N classification - if incomplete data this reviewer suggests to remove the subjects from that center from the study unless there is a strong rationale to as why to keep them in.

- The authors should address the p-tau cut-off employed in Antwerpen if it is not a typographical error

- More information about the controls should be provided. It is widely known that many cognitive controls exhibit altered CSF AD biomarkers with no impact on clinical parameters - the authors need to discuss this aspect 

- Included subjects are described as Caucasian, the authors need to state whether ethnicity was self-reported by the study subjects or assessed by the examining physician only by the appearance of skin color. By tradition ethnicity has not been routinely logged in some European countries (contrary to for example the USA) - since the authors explicitly state that the study was performed on Caucasians they need to comment on this assessment. 

- The authors do not give a proper rationale to as why non-Caucasians were excluded from the study

Minor:

- The authors should explain the A/T/N classification abbreviation in the abstract 

- References are lacking to the methodologies used to assess APOE genotype (M&M)

- The authors need to explain the term 'CSF AD cases'

- There are several language errors (typos) which need to be corrected 

- The statistical method used to assess group differences should be described in the text and figure legends

*** Reviewer #3: 

In this manuscript Saddiki and colleagues use the rigorous A/T/N framework to re-evaluate risk of AD and APOE genotype. They use a case control design (1,599 cases and 801 controls and 11,724 controls) and calculated odds ratios and population attributable fraction (PAF) for AD association with APOE genotypes and age. They report OR of having AD is 4.6 with one E4 copy and 25.4 with two copies. They report highest PAF (69.7) between ages 65-70, and significantly weaker PAF at the tails of the age distributions. They find a modest effect of sex with females showing a stronger effect of E4 on AD risk. They conclude that E4 confers even greater risk that previously appreciated in a cohort rigorously defined using A/T/N biomarker criteria 

This is a nice study in which the authors make a strong case for the use of the A/T/N framework in order to reduce misclassification noise from clinical diagnosis. They show that APOE effects on AD risk may be stronger than previously known, vary across aging, and disproportionately affect women. This is an important study and substantially contributes to our understanding of the two most well-known risk factors for AD- age and APOE. The strengths of the study include biomarker-based diagnosis rather than clinical diagnosis for the primary contrasts (CSF AD vs CSF controls) and the relatively large sample size of all participant groups. The comparison of the CSF AD and the population control cohort is really an extension of the CSF AD vs CSF control group since the population cohort was not defined using the A/T/N criteria. However, this is not really a weakness or significant limitation, but inferences about the CSF AD vs population controls should be made with this caveat. A more concerning limitation is the exclusion of non-Caucasian participants. The risk for AD is different for Caucasian and non-Caucasians and this is something that can be addressed in future studies. 

I have a few additional comments:

1) Because participants were all Caucasian, other risk factors for AD which are over represented in non-Caucasian populations including vascular and metabolic factors may be under represented here. APOE risk for AD may be overstated in this study because these other risk factors were underrepresented. The authors should state this as a limitation of the study, or alternately qualify all of the findings as applying to Caucasians.

2) Its misleading to call the comparison between the CSF AD and CSF controls "secondary analyses" (line 166-168. The "cleaner" control group here is the CSF controls because they were identified by the more rigorous criteria (A/T/N). Further, its not entirely clear how the CSF controls were recruited and under what conditions. The authors state they came from the memory centers (line 168) but other than stating they were required to have normal A/T/N values its not clear who these participants were. Were non-Caucasian excluded from this group? Were they spouses or siblings of the CSF AD participants? A little more detail on how these CSF controls were included in the study would be helpful to understand whether they are comparable to the population control group. 

3) The authors might consider minimizing the notion that they "…ensured that the controls didn't include very early stage of dementia" (lines 347-349). To be fair, they did exclude prevalent and incident AD in their population controls through clinical evaluations which does minimize misclassification, but a major premise of this study is that biomarker diagnosis is superior to clinical diagnosis through reducing misclassification bias. Only the 801 CSF control participants met this scrutiny, the 11,724 population controls did not. 

4) Table 1 presents a single p value for what is presumably a students t-test for the continuous variables age and MMSE, but there are three groups (CSF AD, CSF controls, population controls). The authors need to clarify which comparison the p value reflects or provide p values for all group comparisons. 

4) Sentences on Lines 123-125 and 349-352 are awkwardly written.

***

[LINK]

---

## [Decision Letter · Decision Letter 1]

20 May 2020

Dear Dr. Dumurgier,

Thank you very much for submitting your revised manuscript "Age and the association between apolipoprotein E genotype and Alzheimer’s disease: a cerebrospinal fluid biomarker-based case-control study." (PMEDICINE-D-19-04199R1) for consideration at PLOS Medicine. 

Your paper was re-evaluated by a senior editor and discussed among all the editors here. It was also sent back to two of the original reviewers, and was seen by a statistical reviewer. We apologize for the delayed decision on your revised submission, and will ensure that subsequent reviews will be faster. The original statistical reviewer (Reviewer 1) was not available to re-review the revised version of the manuscript. Therefore, we recruited an additional statistical expert to evaluate the study (Reviewer 4). The reviews are appended at the bottom of this email and any accompanying reviewer attachments can be seen via the link below:

[LINK]

In light of these reviews, I am afraid that we will not be able to accept the manuscript for publication in the journal in its current form, but we would like to consider a revised version that addresses the reviewers' and editors' comments. Obviously we cannot make any decision about publication until we have seen the revised manuscript and your response, and we plan to seek re-review by Reviewer 4. In your revision, please do address the comments of the statistical reviewer (Reviewer 4), particularly their question regarding the use of covariates vs. propensity score matching (also brought up initially by Reviewer 1, point 3), and their question about why age was analyzed both as a categorized and continuous variable.

We expect to receive your revised manuscript by May 27 2020 11:59PM. Please email us (plosmedicine@plos.org) if you have any questions or concerns.

We look forward to receiving your revised manuscript. 

Sincerely,

Caitlin Moyer, Ph.D.

Associate Editor 

PLOS Medicine

plosmedicine.org

1. Title: Please include the study population in the title.

2. Abstract (and throughout): Please replace "Caucasian" with "white" throughout the paper.

3. Table 1: Please define abbreviation for MMSE in the legend.

4. S1 Text: Thank you for including your analysis plan. You note that: “The analyses undertaken were proposed as an MSc internship project advertised via various university channels in November 2018.” Is there any version of the document that incorporates this explanation, or the date, to note that these analyses were pre-specified?

Comments from the reviewers:

Reviewer #2: The authors have nicely addressed my comments and those of the other reviewers. I have no further concerns. 

Reviewer #3: In this revised version of their manuscript, Saddiki and colleagues have made changes that address most of the reviewers concerns. I think this is a strong study which will contribute to the field. The major limitation in my mind is still the restriction to Caucasian participants. Future studies will need to address this finding in more diverse populations.

Reviewer #4: I confine my remarks to statistical aspects of this paper.

The general approach is fine. I do have a couple of questions and issues to resolve before I can recommend publication.

My main question is why the authors chose to add covariates rather than do some form of matching (either by particular values of variables or by propensity score). While it isn't necessarily wrong to use covariates, I think matching is more common.

Lines 274-279 I am confused as to why both these were done - I would not have done the analysis with categorized age. Categorizing independent variables is almost always a mistake. But then the authors also did he analysis with age as continuous and added a quadratic term. So, what was the first analysis for? (Also, the authors could explore use of a spline method and see if it does much better than just the quardratic).

Figs. 2 and 3 - Stacked bar charts are not a good method (see the work of William S. Cleveland) It would be better to use a line graph with 3 lines in each panel of fig 2 and 2 lines in fig. 3. 

Peter Flom

[LINK]

---

## [Decision Letter · Decision Letter 2]

2 Jul 2020

Dear Dr. Dumurgier,

Thank you very much for re-submitting your manuscript "Age and the association between Apolipoprotein E genotype and Alzheimer’s disease: a cerebrospinal fluid biomarker-based case-control study on white older adults." (PMEDICINE-D-19-04199R2) for review by PLOS Medicine.

I have discussed the paper with my colleagues and it was also seen again by one reviewer. I am pleased to say that provided the remaining editorial and production issues are dealt with we are planning to accept the paper for publication in the journal.

[LINK]

We look forward to receiving the revised manuscript by Jul 09 2020 11:59PM. 

Sincerely,

Caitlin Moyer, Ph.D.

Associate Editor 

PLOS Medicine

plosmedicine.org

Requests from Editors:

1.Title: We appreciate your willingness to revise your title; please remove “on white older adults” from the title.

2.Data availability statement: The Data Availability Statement (DAS) requires revision. For each data source used in your study: if the data are owned by a third party but freely available upon request, please note this and state the owner of the data set and contact information for data requests (web or email address). Note that a study author cannot be the contact person for the data.

Please revise your statement to include the relevant contact information for requesting data access for the ADNI Study, 3-City Study and Whitehall II Study, and please note that the contact cannot be one of the authors of the manuscript.

Please also update the link for the Whitehall-II study, as it does not seem to work: https://www.ucl.ac.uk/epidemiology-health-care/research/epidemiology-and-publichealth/research/whitehall-ii

3. Abstract: Line 84: Please replace "subject" with participant, patient, individual, or person.

4. Abstract: Conclusions: Please revise the first sentence to better reflect on the study’s findings: “In this study, we found that AD was associated with APOE ε4 carrier status, with a higher OR than previously reported for associations between AD diagnosis and biomarker status.” or similar.

5. Author summary: “What did the researchers do and find?”: We suggest removing the word “strongly” from the third bullet point, and replacing with “significantly”

6. Author summary: “What do these findings mean?”: We suggest revising the first bullet point to: “Incorporating biomarkers for diagnosis of AD identified an association with APOE4 that is apparently greater than has been previously reported using clinical diagnosis of the disease.” or similar, as the objective of the study does not seem to be to compare associations between this and other studies.

7. Introduction: Line 163: Please replace "subject" with participant, patient, individual, or person.

8. Methods: Line 194: “Term” should be “terms”

9. Results: Line 317: Please revise this sentence to “The proportion of APOE ε4 carriers…”

10. Results: Line 366: Please revise this sentence to “There was no evidence that the age-related association of APOE ε4 differed by sex…”

11. Discussion: Line 434 and 438: Please spell out the word “two” rather than “2” here.

12. Discussion: Line 447: Should “biomarkers” be “biomarker” here?

13. Discussion: Line 452: Please revise to avoid implying causality“...AD is strongly dependent on age at diagnosis.” to “...AD is mediated by age at diagnosis.” In addition, would it be more appropriate to delete “at diagnosis” because your study only indirectly associated age at CSF study with time of diagnosis?

14. Discussion: Line 457 and 460: Why is “AD” in quotations here? Please remove the quotes if not necessary.

15. Discussion: Conclusions paragraph: Please revise the first sentence of the conclusions paragraph to better reflect your study’s findings. “This study supports an association between APOE ε4 and the risk of AD evaluated using biomarkers of AD neuropathology, and the results suggest that the strength of this association varies by age.”

16. Figure 1: Please change “allele” to “alleles”. Please change the X axis to read “Age at CSF testing” or similar. In the legend, please explain the dashed horizontal line at 50.

17. Figure 2: Please provide an X axis label.

18. Figure 6: Please include an X axis label indicating the values are OR and 95% CIs.

19. Supporting information Figure 1: In the title, there should be a space between CSF and AD.

20. Supplementary table 1: Is it possible to elaborate in the legend, or use a more specific term than “cut-off” as it isn’t quite clear exactly what it means.

Comments from Reviewers:

Reviewer #4: The authors have addressed my concerns and I now recommend publication

Peter Flom

[LINK]

---

## [Editor Report · Decision Letter 3]

22 Jul 2020

Dear Dr Dumurgier, 

On behalf of my colleagues and the academic editor, Dr. Raquel C. Gardner, I am delighted to inform you that your manuscript entitled "Age and the association between Apolipoprotein E genotype and Alzheimer’s disease: a cerebrospinal fluid biomarker-based case-control study." (PMEDICINE-D-19-04199R3) has been accepted for publication in PLOS Medicine. 

PRODUCTION PROCESS

PRESS

PROFILE INFORMATION

Thank you again for submitting the manuscript to PLOS Medicine. We look forward to publishing it. 

Best wishes, 

Caitlin Moyer, Ph.D.

Associate Editor 

PLOS Medicine

plosmedicine.org